# Research on Liquid Flow Measurement Method Based on Heat Transfer Method

**Hongwei Qin, Ruirong Dang * and Bo Dang**

Shaanxi Provincial Key Laboratory of Oil and Gas Well Measurement and Control Technology,
Xi'an Shiyou University, Xi'an 710065, China
* Correspondence: dangrr@xsyu.edu.cn

**Abstract:** Thermal flowmeters are used more and more widely in liquid flow measurement. In this paper, the mechanical shape of the thermal flowmeter is designed, and the optimal installation position of the thermal probe is determined. In the aspect of measurement mechanism research, three heating methods of the thermal probe are deduced: constant voltage heating method, constant current heating method, and constant power heating method. After reasoning, the constant current heating method is determined to be ideal, so the constant current heating method is selected to heat the speed-measuring probe in the experiment. By analyzing the power factor of convection heat transfer and residual heat source of the heating probe, it is concluded that the measurement range of the thermal flowmeter is 0.5–15 $m^3/d$, the flow in this range is proportional to the electrical signal, and the relative error of measurement is within ±5.8%. According to the analysis of the experimental results, the thermal flowmeter has a simple mechanical structure and no redundant moving parts, which can prolong its service life when used on site. When considering industrial applications, the error may be greater than the laboratory error.

**Keywords:** heat transfer method; thermal flowmeter; constant current method; power factor

## 1. Introduction

A thermal flowmeter is mainly a tool for indirectly calculating the flow of heat in fluid heat conduction, convection heat transfer, heat radiation, and other transfer modes through the heat transfer method [1–3]. The idea of thermal flow measurement was first proposed by Thomas of the United States at the beginning of the 20th century [4]. The development of thermal flowmeter is widely used in flow measurement at home and abroad [5]. The development of thermal flowmeter is widely used in flow measurement, which has been studied in gas and liquid measurement [6]. In gas flow measurement, the development of thermal flowmeter is relatively mature, and there are a lot of practical applications at home and abroad [7,8]. In the gas experiment, keeping the specific heat of the gas unchanged, the amount of gas is calculated by measuring the consumed electric energy under the condition of maintaining a small temperature difference [9]. The gas thermal flowmeter is relatively mature in the market. In this paper, a liquid mass flowmeter is developed according to the measuring principle of the gas thermal flowmeter. However, the development of liquid thermal flowmeter is relatively sluggish, and international scholars have studied more convection heat transfer [10,11] and flow around a cylinder [12–14]. The thermal fluid flowmeter in China is still in the research and development stage, and many difficulties need to be overcome.

Around 1990, Huijsing, J.H et al. [15] designed a new thermal mass flowmeter for measuring fuel consumption of automobiles and light aircraft and liquid flow in industrial processes. China successively carried out research on thermal flowmeters and made some progress.

Liu, D et al. [16] carried out a flow measurement experiment of constant flow heating of oil–water two-phase flow with different water cuts in vertical simulation wells. The

experimental results show that the platinum resistance decreases more with the increase of the flow rate. The results show that the resolution is good in the flow range of 1–40 $m^3/d$, and the measurement error of this thermal flowmeter is smaller than the standard value under the condition of ultra-high water cut. When the water cut is above 90%, the measurement error is within 8%, and when the water cut is about 80%, the error is within 25.9%. It can be seen that the error increases as the water cut decreases. Jeong, U. [17] proposed to use PMPF to obtain liquid sodium flow in a wide temperature range. A non-stationary method was adopted for the calibration of the probe given the liquid sodium temperature range of 150–415 °C. A relationship between the measured voltage signal and the flow rate was obtained successfully. One of the main problems in reference [18] is how to determine the water density in the calibration facility under actual conditions or reference actual conditions and derive approximate functions describing the temperature characteristics of water density, which are applicable to uncertainty analysis. Doh, I. [19] proposed a miniaturized thermal flowmeter consisting of a silicon substrate, a platinum heater layer on a silicon dioxide thin membrane, and a polymer microchannel to provide accurate flow-rate measurement. The present thermal flowmeter is fabricated by the micromachining and micromolding process and exhibits sensitivity, linearity, and uncertainty of 0.722 mW/(g/h), 98.7%, and (2.36 ± 0.80)%, respectively, in the flow-rate range of 0.5–2.5 g/h when the flowmeter is operated in the constant temperature mode with the channel width of 0.5 mm. The measurement range of the flow rate can be easily adjusted by changing the cross-sectional microchannel dimension. The present miniaturized thermal flowmeter shows a high potential for infusion-pump calibration in clinical settings. Yang, Y. et al. [20] developed a constant power thermal flowmeter for liquid phase flow and used Matlab numerical simulation software to analyze the relationship between the measurement results of a thermal flowmeter and oil–water two-phase flow under different water holdup conditions. The results show that the output voltage of the thermal flowmeter decreases monotonously with the increase of water holdup at the same flow rate, so the flow measurement of the thermal flowmeter must be corrected in combination with the water cut.

It is also applied in gas flow measurement. Rupnik, K. et al. [21] proposed a new measurement method to identify the gas type in the thermal dispersion mass flowmeter. First, a temperature flow model was established, and flow measurement experiments were carried out on five different gases under the condition of maintaining a constant temperature difference. The results show the relationship between $P/\Delta T$ and flow, and $P/\Delta T$ increases with the increase of flow. In addition, the height of the power curve is different, indicating that different gases require different power to maintain constant temperature differences under the same flow rate. Therefore, further correction work is needed to lay a foundation for the subsequent phase of content detection. Bekraoui, A. et al. [22] designed the structure of the thermal gas flowmeter, analyzed the measurement principle of the thermal flowmeter using the energy balance formula, and deduced the formula of the hot wire anemometer. A constant temperature hot wire system was established, the theoretical formula was derived, and simulation verification was made using Matlab. Based on the industrial data of a natural gas field, the influence of fluid velocity on the resistance value was explored. The experimental results showed that the resistance value of the velocity probe decreased with the increase in flow rate, and the slope was large in the low-speed area. The heating power increased with the increase in flow rate, and the increase was large in the low-speed area. The conclusion that the practice is consistent with the theory is drawn. Bekraoui, A. et al. [23] studied the heat transfer formula of the hot wire anemometer and obtained the functional relationship between constant power heating and gas flow. Yu, Y.M et al. [24] provided different constant power to the heater to realize the flow detection experiment of liquid $CO_2$. The experimental results show that the temperature difference when the flow is lower $\Delta T$ is higher; the higher the added power, the higher the overall curve.

It can be seen from the previous review that the research on gas thermal flowmeters is relatively mature, but the research on liquid thermal flowmeters is relatively few. Therefore,

this paper improves the mechanical structure on the basis of reference [15] and analyzes the measurement mechanism of the liquid flowmeter through the comparison of three heating methods to improve the resolution and linearity of the thermal flowmeter. The thermal flowmeter in this paper is mainly used in a single-phase fluid. In order to further improve the utilization value, this thermal flowmeter is integrated with other multi-sensor technologies for mutual correction and compensation. This will provide a basis for testing and monitoring oil-water mixed flow and phase holdup of ultra-high water content wells in the oilfield in the later period.

## 2. Study on Measuring Mechanism of Thermal Flowmeter

It is very important to study the measurement mechanism of thermal flowmeter based on heat transfer method. Mechanical structure design requires high response speed of measurement system. The correct selection of heating mode of thermal probe can improve the measurement accuracy.

### 2.1. Mechanical Structure Design of Flowmeter

The external structure of the thermal flowmeter is shown in Figure 1. The external structure is mainly composed of the meter head, pipe, and thermal probe. In the design, the thermal probe shall be placed inside the pipe. The inner diameter of the pipe is 50 mm, the distance between probes is 80 mm, the length of each probe is 40 mm, and the diameter of the probe is 3 mm. The temperature measuring probe is PT1000 platinum resistance, and the heating probe is PT20 platinum resistance. In the experimental design, the temperature-measuring probe is placed at the inlet of the pipe, and the velocity-measuring probe is placed at the outlet of the pipe. Each thermal probe shall be installed perpendicular to the pipe. The thermal flowmeter works on the principle of electric heating, so the installation structure must ensure that all thermal probes are immersed in the fluid; otherwise, the probe may be burned by heat. Thermal probes are considered to have an axisymmetric internal structure with defined dimensions and material properties.

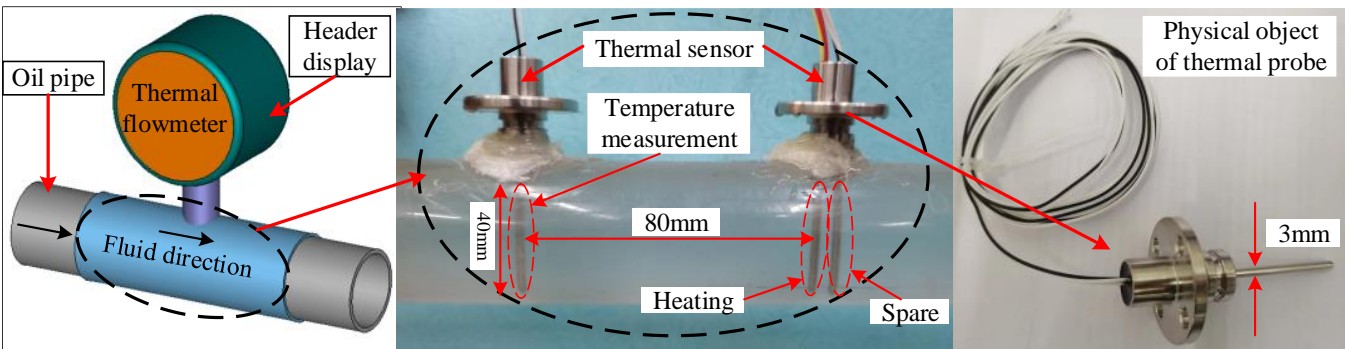

**Figure 1.** Mechanical structure of thermal flowmeter.

### 2.2. Constant Voltage Heating

The constant voltage heating method is to add a constant voltage $V_c$ to the total resistance. Figure 2 shows the constant voltage heating mode. $R_1$ is a power resistor with constant resistance, and $R$ is a PT20 platinum resistor with variable resistance. Throughout the experiment, $R_w$ is a measurable variable.

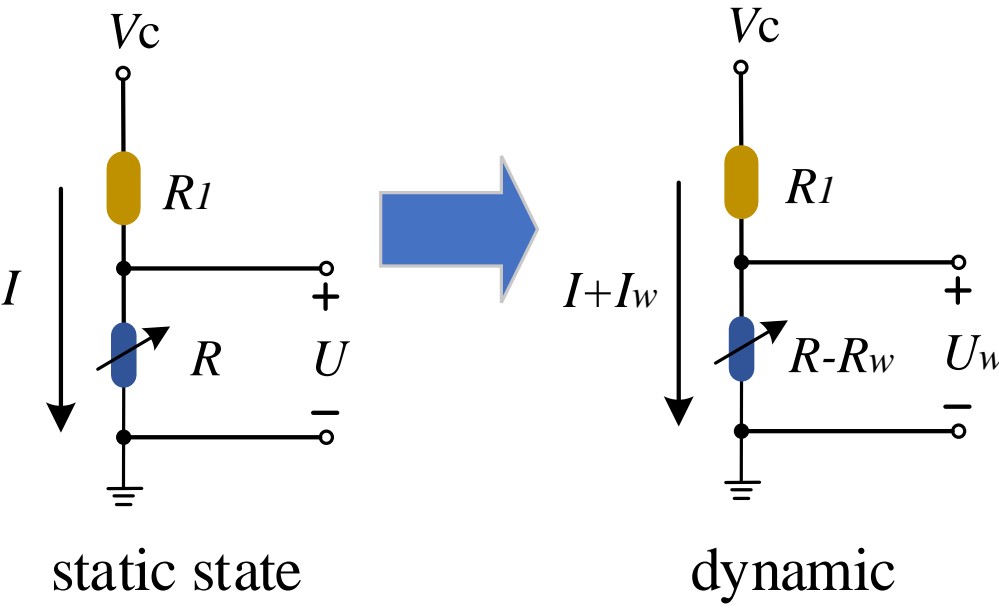

**Figure 2.** Constant voltage heating.

When the fluid is stationary, the equilibrium formula can be expressed as Formula (1). The relationship between the resistance value $R$ of the platinum resistance and the temperature $T$ can be expressed as Formula (2).

$$U = \frac{R}{R_1 + R} V_c \tag{1}$$

$$R = R_0(1 + aT + bT^2) \tag{2}$$

In (2), $R_0$ is the resistance value of PT20 at $0\,^\circ$C, and $a$ and $b$ are proportional coefficients. The coefficient $b$ has a small order of magnitude. In this paper, ignoring the higher order term of $b$, we can get:

$$R = R_0(1 + aT) \tag{3}$$

The simultaneous Formulas (1) and (3) can obtain:

$$U = \frac{V_c(R_0 + aR_0 T)}{R_1 + R_0 + aR_0 T} \tag{4}$$

It can be seen from Formula (4) that in a static fluid, when the fluid temperature is constant, the current will not change. The voltage $U$ at both ends of the platinum resistance is not linear with the change of fluid temperature $T$. According to the change results shown in Figure 2, when the fluid flows, the resistance of the platinum resistance decreases and the current passing through the platinum resistance increases. Therefore, the voltage $U_w$ at both ends of the platinum resistance during fluid dynamics can be expressed as:

$$U_w = \frac{V_c(R_0 + aR_0 T - R_w)}{R_1 + R_0 + aR_0 T - R_w} \tag{5}$$

$U_w$ in Formula (5) changes with $T$, and other parameters are known or measurable. So, the differential form of $U_w$ and $T$ in formula (5) can be expressed as:

$$\frac{dU_w}{dT} = \frac{aV_c R_0 R_1}{(R_1 + R_0 + aR_0 T - R_w)^2} \tag{6}$$

It can be seen from Formula (6) that the change of $U_w$ is no longer linear with temperature $T$.

### 2.3. Constant Current Heating

The constant current heating method is to add a constant current source at both ends of the platinum resistor. The heat taken away by the platinum resistance will cause the resistance value of the platinum resistance to change when the flow rate of the fluid changes. The relationship between voltage and temperature can be obtained by measuring the voltage at both ends of the platinum resistance. Constant current heating mode is shown in Figure 3.

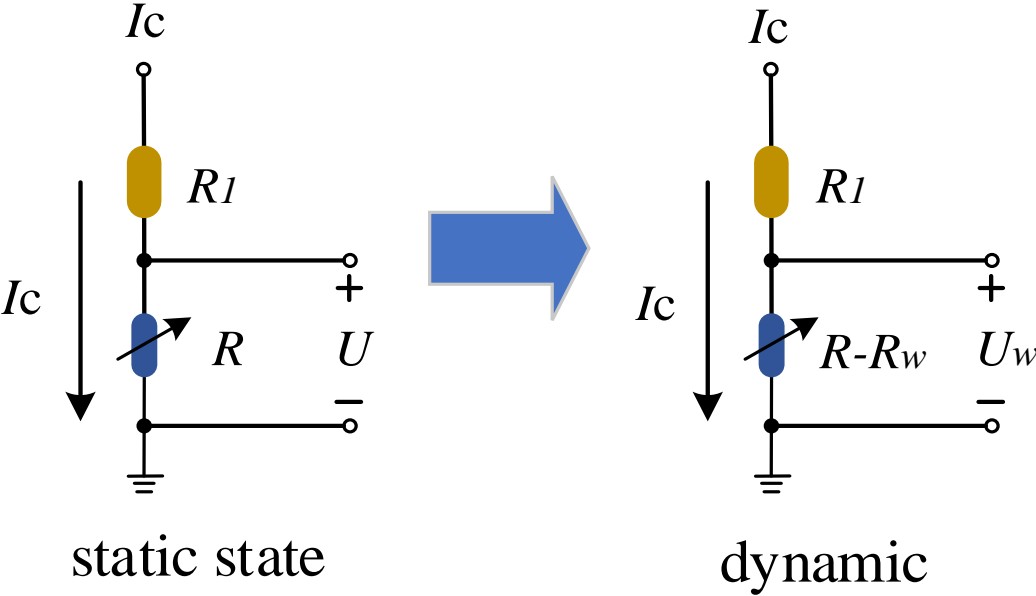

**Figure 3.** Constant current heating.

When the fluid is still, constant current heating mode is adopted, and the relationship between platinum resistance and voltage is expressed as Formula (7). The relationship of Formula (8) can be obtained by bringing Formula (3) into Formula (7).

$$U = I_c R \tag{7}$$

$$U = a I_c R_0 T + I_c R_0 \tag{8}$$

When the fluid flows, the relationship between the platinum resistance value and the voltage is expressed as formula (9).

$$U_w = a I_c R_0 T + (I_c R_0 - I_c R_w) \tag{9}$$

It can be seen from Formula (9) that under the condition of constant current, the voltage at both ends of the platinum resistance always keeps a positive proportion to the temperature of the platinum resistance. As the flow rate changes with time, $U_w$ in Formula (9) changes with $T$, and other parameters are known or measurable. So, the differential form of $U_w$ and $T$ in Formula (10) can be expressed as follows:

$$\frac{dU_w}{dT} = a I_c R_0 \tag{10}$$

From the transient change of Formula (10), it can be seen that the voltage $U_w$ at both ends of the platinum resistance always keep a linear relationship with the temperature $T$ of the platinum resistance, where $R_w$ can be measured. This linear relationship is conducive to subsequent experimental calibration and measurement, etc.

### 2.4. Constant Power Heating

Constant power heating is to add constant power to both ends of the platinum resistance, measure the voltage value at both ends of the platinum resistance when the power is unchanged, and deduce the relationship between voltage and flow through the relationship between voltage and temperature. Constant power heating mode is shown in Figure 4.

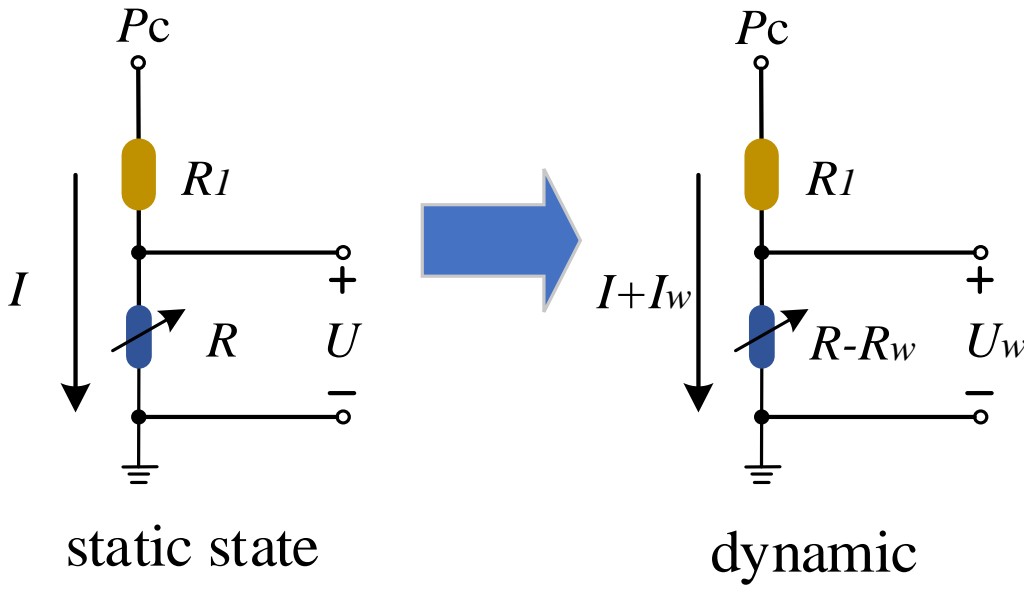

**Figure 4.** Constant power heating.

In Figure 4, when the fluid is still in the constant power heating mode, the relationship between the platinum resistance value and the voltage is shown as follows:

$$U = (\frac{P_c}{R_1 + R})^{\frac{1}{2}} R \tag{11}$$

The relationship between the voltage and temperature when Formula (3) is introduced into Formula (11) is as follows:

$$U = (\frac{P_c}{R_1 + R_0 + aR_0T})^{\frac{1}{2}} (R_0 + aR_0T) \tag{12}$$

When the fluid flows, the relationship between the platinum resistance value and the voltage is expressed as formula (13).

$$U_w = (\frac{P_c}{R_1 + R_0 + aR_0T - R_w})^{\frac{1}{2}} (R_0 + aR_0T - R_w) \tag{13}$$

As the flow rate changes with time, $U_w$ in Formula (13) changes with $T$, and other parameters are known or measurable. So, the differential form of $U_w$ and $T$ in Formula (14) can be expressed as follows:

$$\frac{dU_w}{dT} = \frac{1}{2} \left( \frac{P_c}{R_1 + R_0 + aR_0T - R_w} \right)^{-\frac{1}{2}} \frac{-aR_0P_c}{(R_1 + R_0 + aR_0T - R_w)^2} (R_0 + aR_0T - R_w)$$
$$-aR_0 \left( \frac{P_c}{R_1 + R_0 + aR_0T - R_w} \right)^{\frac{1}{2}} \tag{14}$$

According to Formula (14), the relationship between $U_w$ at both ends of the platinum resistance and temperature $T$ is not linear with time, and the current and voltage at both ends of the platinum resistance change with time at the same time.

To sum up the comparison of the above three methods, the linear relationship between $U_w$ at both ends of the platinum resistance and temperature $T$ with time synchronization can only be obtained by constant current heating. The temperature of the speed measuring probe is linear with the size of the flow rate, so it can be known that $U_w$ at both ends of the platinum resistance is proportional to the flow rate. Compared with the constant power heating method in reference [15], the constant current heating method in this paper improves the linearity and resolution.

## 3. Discussion on Power Factor of Constant Current Heating

When the fluid flows, the heat $E$ taken away from the surface of the heating source by forced convection heat transfer can be expressed by the Newton cooling formula as follows:

$$E = hA(t_h - t_e) \tag{15}$$

In Formula (15), $E$ represents forced convection heat transfer, W; $h$ is the heat transfer coefficient, W/m$^2$/°C; $A$ is the surface area of the heating probe, m$^2$; $t_h$, $t_e$ represents the temperature of the heating probe and the temperature of the background where the fluid is located, °C. The surface area of the heating probe can be expressed as (16), where $l$ is the probe length, m, and $d$ is the probe diameter, m.

$$A = \pi l d \tag{16}$$

In order to clearly analyze the flow characteristics, the thermodynamic parameters Nussel number ($N_\mu$), Prandtl number ($P_r$), and Reynolds number ($R_e$) are introduced in this paper to express the functional relationship between heating power and fluid heat transfer. The expression is as follows:

$$N_\mu = \frac{hD}{\lambda_f} \tag{17}$$

$$P_r = \frac{\eta C_p}{\lambda_f} \tag{18}$$

$$R_e = \frac{\rho v D}{\eta} \tag{19}$$

In Formulas (17)–(19), $\lambda_f$ is the thermal conductivity of the fluid, W/m/°C; $\eta$ is the dynamic viscosity of the fluid, $Pa\cdot s$; $C_p$ is the specific heat capacity of fluid at constant pressure; $\rho$ is the fluid density, kg/m$^3$; $v$ is the fluid velocity, m/s; and $D$ is the diameter of fluid pipe, m. When the measured fluid fills the pipe, the interference of natural convection is ignored. At this time, the heat $E$ of the forced convection heat transfer zone of the thermal probe obtained from the simultaneous Formulas (15)–(17) can be expressed as follows:

$$E = \pi l d \left(\lambda_f N_u / D\right)(t_h - t_e) \tag{20}$$

According to the convection heat transfer formula given by C.C. Thomas [4], under certain conditions, $N_u$ can be expressed as follows:

$$N_u = 0.42 P_r^{0.2} + 0.57 P_r^{0.33} R_e^m \tag{21}$$

The exponent of Reynolds number $R_e$ in Formula (21) will change with the change of flow velocity, and it is reasonable to take 0.4 as $m$ after many experiments. The simultaneous Formulas (16)–(21) can get the expression of heat $E$ in the convection zone:

$$E = (\pi l d \lambda_f / D)(t_h - t_e)[0.42 P_r^{0.2} + 0.57 P_r^{0.33}(\rho D / \eta)^m v^m] \tag{22}$$

Using Formulas (23) and (24) to express $A_c$ and $B_c$ instead of the parameters in Formula (22).

$$A_c = 0.42 \pi l (d/D) \lambda_f (P_r)^{0.2} \tag{23}$$

$$B_c = 0.57 \pi l (d/D) \lambda_f P_r^{0.33}(\rho D / \eta)^m \tag{24}$$

Therefore, Formula (22) can be simplified:

$$E = (A_c + B_c v^m)(t_h - t_e) \tag{25}$$

The residual heat of heat source $H$ can be expressed:

$$H = P - E \tag{26}$$

The proportional relationship between heating power and residual heat can be expressed:

$$P = kH \tag{27}$$

In Formula (27), $P$ represents the power of the heater, which can be measured through the experimental process. At the same time, the voltage value at both ends of the platinum resistance can be measured, and the fluid flow rate can be measured according to the voltage value. In Formula (27), $k$ represents the power factor of the heater. The relationship between the flow rate and the probe heat can be judged by calculating the power factor.

## 4. Experimental Analysis and Discussion

### 4.1. Introduction to Experimental Environment

This experiment was completed in the Key Laboratory of Oil and Gas Well Measurement and Control in Shaanxi Province of Xi'an Shiyou University. Figure 5 shows the experimental environment, mainly including circulating pump, fluid pipeline, flow regulating valve, water storage tank, DC regulated power supply, high-precision ammeter, fluid outlet valve, exhaust port, thermal probe, related hardware circuit, and PC.

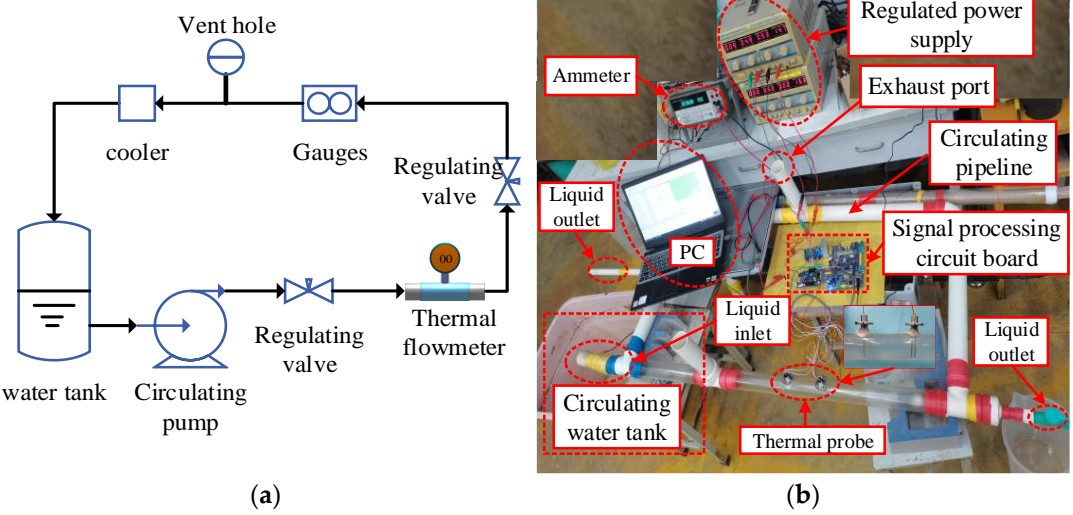

**Figure 5.** Experimental environment. (**a**) Schematic diagram of experimental circulation system. (**b**) Construction chart of experimental circulation system.

During the experiment, there is a circulating pump with controllable flow rate in the circulating water tank. The fluid will discharge bubbles from the liquid through the pipeline to ensure that the fluid fills the pipeline and eliminate the interference of bubbles to the measurement. The background temperature is measured by the temperature measuring probe, and then the fluid velocity is measured by the velocity measuring probe. For fluid background temperature measurement, a small current of 1 mA is required to drive the temperature measuring probe. If the current is too high, the temperature measuring probe will heat up. If the current is too low, the driving ability will be too weak. Therefore, it is reasonable to select a current of 1 mA. Considering that the temperature of the velocity probe is much higher than the fluid temperature, it was decided to select 500 mA heating velocity probe after many tests. If the current of the velocity probe is too large, the life of the velocity probe will be reduced, and if the current is too small, the flow measurement range will be reduced.

According to the current value of the constant current source, the relevant software and hardware circuits are designed to drive the thermal probe. In the process of experimental calibration and testing, soft measurement technology is used to suppress noise and extract useful signals. After multiple calibrations in the experiment, it is necessary to repeatedly test the average value to find out the functional relationship between voltage and flow.

### 4.2. Analysis of Experimental Results

In this experiment, constant current source with good linearity is selected for heating, and the water flow of 0.5–25 m$^3$/d is calibrated. The flow less than 0.5 m$^3$/d will generate vortices, which will make the measurement inaccurate. Therefore, the calibrated flow starts from more than 0.5 m$^3$/d. The calibrated model is used to detect the relationship between flow and voltage. Figure 6 shows the linear relationship curve between flow and voltage. It can be seen from Figure 6 that the voltage decreases gradually with the increase of flow. The linearity of the yellow area in Figure 6 is better, and the flow range is 0.5–15 m$^3$/d. The linearity of the green area is poor, and the flow range is 15–25 m$^3$/d. Therefore, Formula (28) can be used to express the range of 0.5–15 m$^3$/d, and the determination coefficient of fitting is 0.995, and the fitting degree is high.

$$V = -0.03007Q + 12.6944 \tag{28}$$

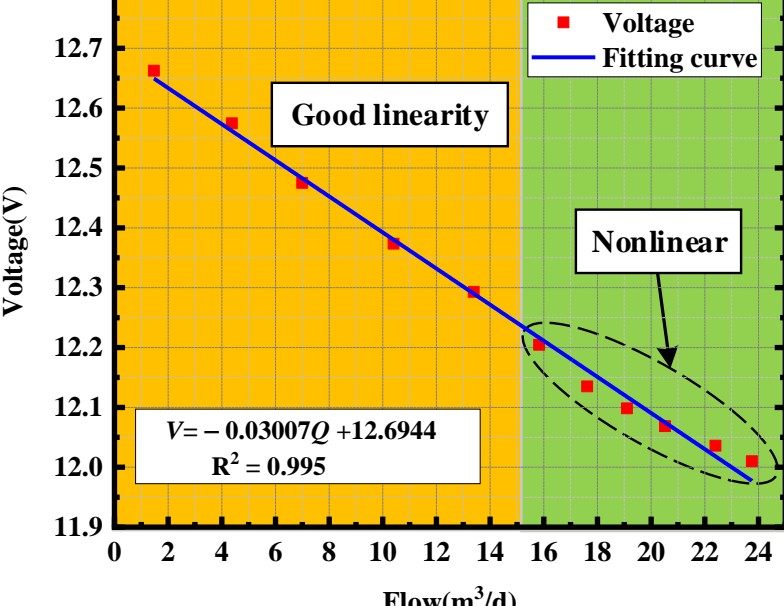

**Figure 6.** Linear diagram of voltage versus flow rate.

In Formula (28), *V* represents voltage value, and *Q* represents flow value.

As shown in Figure 7, the green area with flow range of 15–25 m³/d can be fitted by polynomial, the determination coefficient of fitting is 0.9991, and the fitting degree is high. The fitting function can be expressed by Formula (29).

$$V = 0.00166Q^2 - 0.04151Q + 12.723 \tag{29}$$

Through the relationship between flow conversion and voltage, the relationship between heat transfer coefficient and flow can be obtained. The results are shown in Figure 8. It can be seen that the relationship between the heat transfer coefficient and the flow rate is no longer linear, indicating that the convection heat transfer of the thermal probe will be saturated.

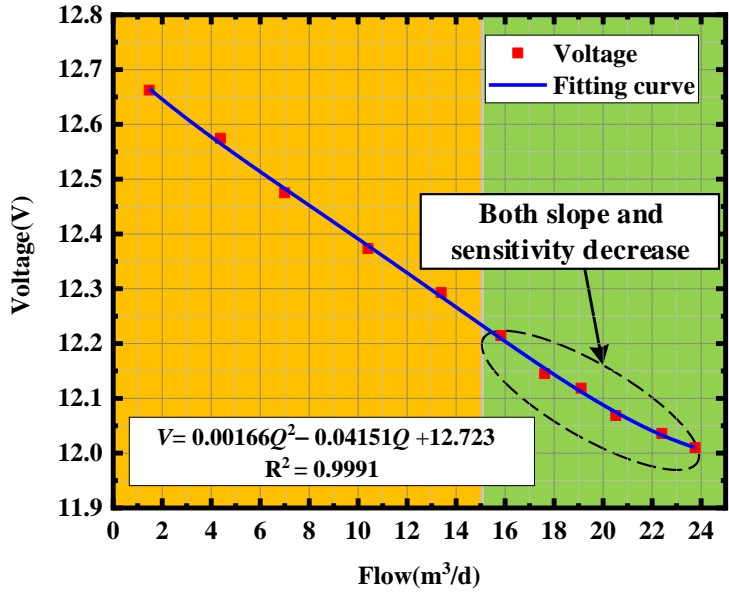

**Figure 7.** The non-linear relationship diagram of voltage with flow.

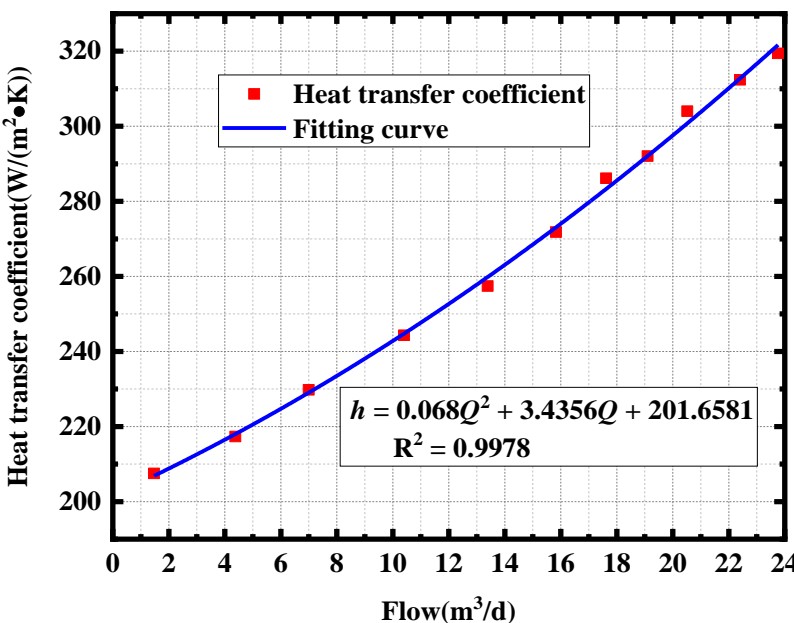

**Figure 8.** Diagram of heat transfer coefficient and flow rate.

The layout of the relationship between power and flow shown in Figure 9 is obtained by Formula (25) and Formula (26). It can be seen from the green area in Figure 9 that with the increase of flow rate, the heat exchange and heat source surplus will no longer increase with the change of flow rate. It can be seen from the calculation results that the calculation of heat exchange power and residual power of heat source is related to the selection of m value in Formula (21). In order to ensure the detection accuracy of low flow rate, the selection of m value should meet the requirements of low flow.

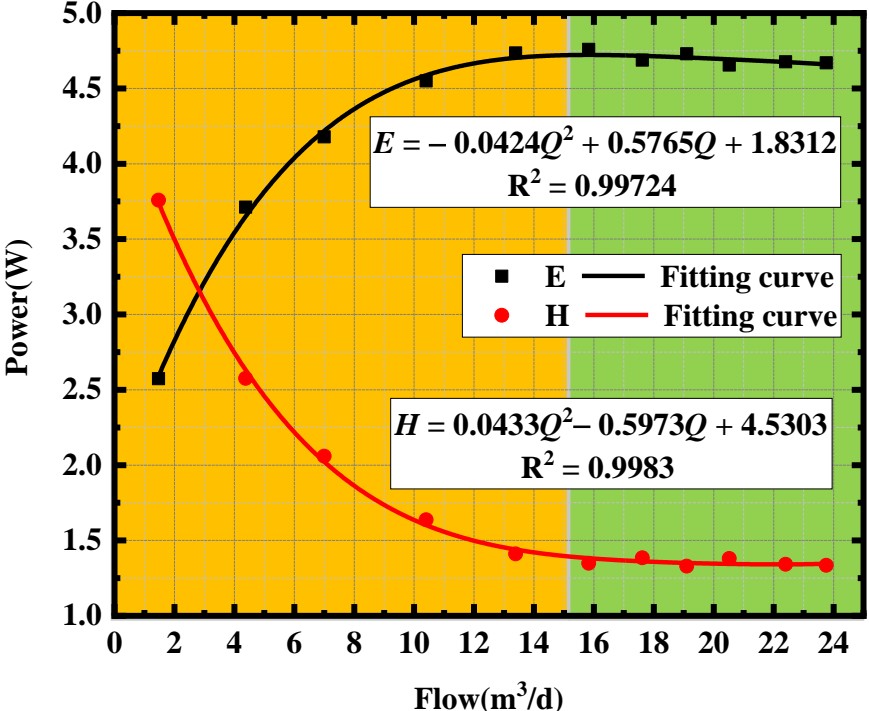

**Figure 9.** Diagram of power and flow.

Figure 10 shows the relationship between the power factor and flow rate. When the flow rate is 15–25 m$^3$/d, the power factor no longer changes significantly. The change of the power factor is the same as the trend of the residual power of the heat source and is also related to the selection of m value in Formula (21). When $m = 0.4$, the linearity of flow and voltage between 0.5–15 m$^3$/d is good. It can be seen from the above analysis that when the flow rate is below 15 m$^3$/d, the linearity between the flow rate and the voltage value at both ends of the platinum resistance is better. Since the thermal probe selected in this paper is commercial on the market, the specifications and parameters of the thermal probe shown cannot be changed. According to the experimental test results, when the flow rate exceeds 15 m$^3$/d, the sensitivity of the thermal probe to the flow rate has reached the saturation state. When the flow rate continues to increase, the sensitivity of the thermal probe becomes poor and the resistance value no longer changes, so the heating power basically remains unchanged.

Through many experimental studies, the measured value is compared with the model calculation value, and it is found that the experimental relative error of 0.5–15 m$^3$/d is within ±5.8%. Because this experiment is tested in a good environment, the relative error is within the acceptable range. However, the actual working conditions will be affected by pipeline friction, fluid salinity, fluid flow pattern, gas, and impurities, and the error will be greater than that of the laboratory. Figure 11 shows the relative error between the measured value and the model calculation value.

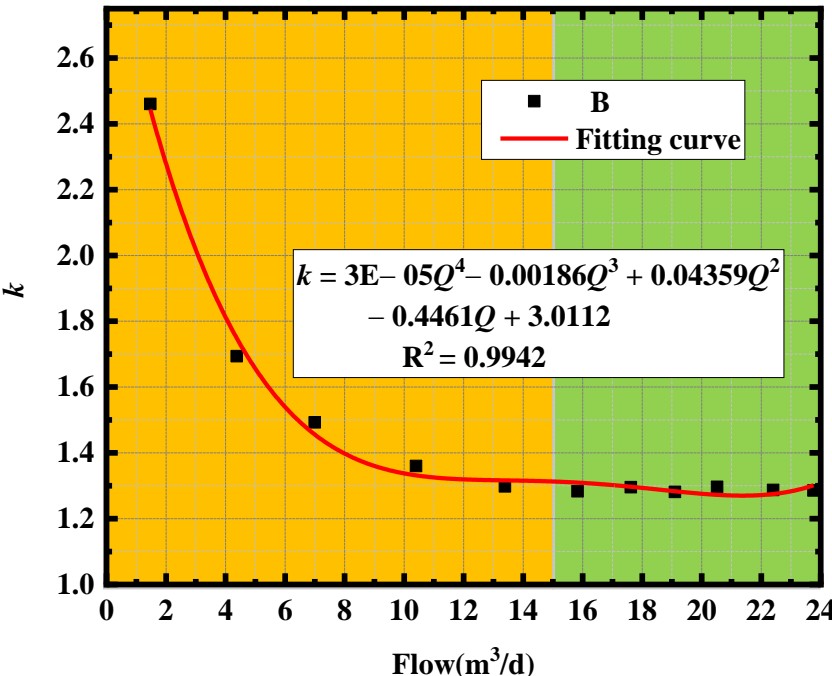

**Figure 10.** Power factor and flow chart.

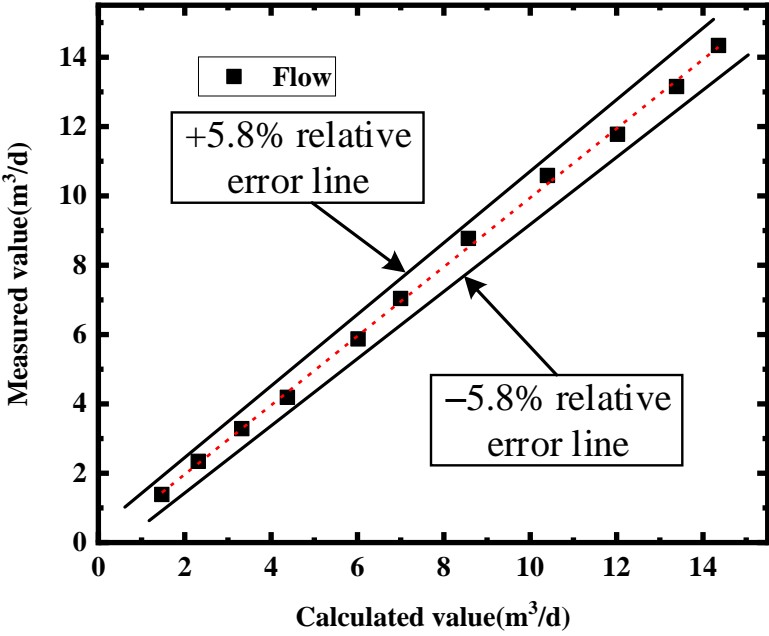

**Figure 11.** Error chart of measured value and calculated value.

## 5. Conclusions

The liquid thermal flowmeter developed in this paper can be measured within 0.5–15 m³/d, and the linearity of flow and voltage is better, so it can be used as the measurement standard of industrial unidirectional fluid. The experimental analysis shows that the relative error of the experiment is ±5.8%. The constant current heating method used in this paper has better linearity and higher resolution than the constant power method in reference [15], which can improve the flow measurement range. This is a major innovation of this paper. The conclusion of this experiment can provide a reliable basis for the flow measurement of ultra-high water cut wells in the oilfield and also lay a foundation for the total flow measurement of multiphase flow.

**Author Contributions:** Conceptualization, H.Q. and B.D.; methodology, H.Q.; software, H.Q.; validation, H.Q. and B.D.; formal analysis, B.D.; investigation, R.D.; resources, R.D.; data curation, H.Q.; writing—original draft preparation, H.Q.; writing—review and editing, H.Q.; visualization, R.D. and B.D.; supervision, R.D.; project administration, R.D.; funding acquisition, R.D. All authors have read and agreed to the published version of the manuscript.

**Funding:** Supported by National Natural Science Foundation of China for online Monitoring of Multi-component Reservoir with TIME-domain electromagnetic Method, approval number: 41874158. Supported by the Postgraduate Innovation and Practice Ability Development Fund of Xi'an Shiyou University, approval number: YCS22211001.

**Institutional Review Board Statement:** Not applicable.

**Informed Consent Statement:** Not applicable.

**Data Availability Statement:** Data available by reasonable request.

**Acknowledgments:** Thank you for the experimental environment provided by Key Laboratory of Oil and Gas Well Measurement and Control in Shaanxi Province of Xi'an shiyou University.

**Conflicts of Interest:** The authors declare that they have no known competing financial interests or personal relationships that could have appeared to influence the work reported in this paper.

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
