# Peer review of "Research on Liquid Flow Measurement Method Based on Heat Transfer Method"

_water, doi:10.3390/w15061052_

Round 1
Reviewer 1 Report
The manuscript Research on Liquid Flow Measurement Method Based on Heat Transfer Method, by Dang et. al is impressive work and I would suggest publishing the article with minor revisions suggested below.
Lines 26-27: The development of thermal flowmeter is widely used in flow measurement, which has been studied in gas and liquid measurement.
Comment: Please add references for the above statements.
Lines 34-36: In the gas experiment, keeping the specific heat of the gas unchanged, the amount of gas is calculated by measuring the consumed electric energy under the condition of maintaining a small temperature difference.
Comment: Reference missing
Comment: Introduction needs more context on the importance of thermal flowmeter. It currently fails to explain why this study of thermal flowmeter is crucial.
A lot of missing references, I would suggest the authors to do a thorough check and add references as needed.
Lines 339-340: Fig 10 shows the relationship between power factor and flow rate. When the flow rate is 15-25m3 /d, the power factor no longer changes significantly.
Comment: Figure 10, at higher flow rate 15-25 m3/d the relationship between the power factor does not appear to be very good fit. Although the R2 value achieved is 0.9942 I think the curve does not fit very well. I would suggest adding a comment why was this observed. Perhaps it might also be important to add what is an acceptable R2 value for all the curves.
Lines 349-350: However, the actual working conditions will be affected by pipeline friction, fluid salinity, fluid flow pattern, gas and impurities, and the error will be greater than that of the laboratory.
Comment: I am failing to understand the importance of this model value if it’s expected to show greater error in actual working condition. I will suggest adding a few advantages of this model even though it’s expected to fail in actual working conditions.
Lines 17-18: According to the analysis of the experimental results, the thermal flowmeter has simple mechanical structure and high accuracy, and can be considered for industrial application.
Comment: This is contradictory to the lines 349-350, where it’s stated that this model will have higher error in actual conditions.
Author Response
The manuscript Research on Liquid Flow Measurement Method Based on Heat Transfer Method, by Dang et. al is impressive work and I would suggest publishing the article with minor revisions suggested below.
Lines 26-27: The development of thermal flowmeter is widely used in flow measurement, which has been studied in gas and liquid measurement.
Comment: Please add references for the above statements.
The author’s answer: The issues in the text have been corrected.
Lines 34-36: In the gas experiment, keeping the specific heat of the gas unchanged, the amount of gas is calculated by measuring the consumed electric energy under the condition of maintaining a small temperature difference.
Comment: Reference missing
Comment: Introduction needs more context on the importance of thermal flowmeter. It currently fails to explain why this study of thermal flowmeter is crucial.
A lot of missing references, I would suggest the authors to do a thorough check and add references as needed.
The author’s answer: The issues in the text have been corrected.
Lines 339-340: Fig 10 shows the relationship between power factor and flow rate. When the flow rate is 15-25m3 /d, the power factor no longer changes significantly.
Comment: Figure 10, at higher flow rate 15-25 m3/d the relationship between the power factor does not appear to be very good fit. Although the R2 value achieved is 0.9942 I think the curve does not fit very well. I would suggest adding a comment why was this observed. Perhaps it might also be important to add what is an acceptable R2 value for all the curves.
The author’s answer: The fitting curve formula in Fig 10 was written incorrectly due to my negligence in the article, and now the error has been corrected. Since the thermal probe selected in this paper is commercial and directly purchased on the market, the specifications and parameters of the thermal probe shown cannot be changed. According to the experimental test results, when the flow rate exceeds 15m3/d, the sensitivity of the thermal probe to the flow rate has reached the saturation state. When the flow rate continues to increase, the sensitivity of the thermal probe becomes poor and the resistance value no longer changes, so the heating power basically remains unchanged.
Lines 349-350: However, the actual working conditions will be affected by pipeline friction, fluid salinity, fluid flow pattern, gas and impurities, and the error will be greater than that of the laboratory.
Comment: I am failing to understand the importance of this model value if it’s expected to show greater error in actual working condition. I will suggest adding a few advantages of this model even though it’s expected to fail in actual working conditions.
The author’s answer: The experimental conditions in the laboratory are particularly ideal. The fluid is tap water, which has eliminated the effects of pipe friction, salinity, impurities, etc., and the gas has been discharged from the exhaust hole. Therefore, the experimental error obtained is relatively small, but the error may increase in actual working conditions. Compared with other flowmeters, the thermal flowmeters designed in this paper have simple mechanical structure and can be used in industrial engineering for a long time. Another advantage is that the thermal probe is not only sensitive to the flow velocity, but also to the fluid composition. Therefore, this paper considers using the same thermal probe to measure multiphase flow in the oil industry in the later stage. What is not clearly stated in the article has been revised and supplemented in detail.
Lines 17-18: According to the analysis of the experimental results, the thermal flowmeter has simple mechanical structure and high accuracy, and can be considered for industrial application.
Comment: This is contradictory to the lines 349-350, where it’s stated that this model will have higher error in actual conditions.
The author’s answer: There are errors in the description of this article, which have been revised and supplemented in detail.

Reviewer 2 Report
Review Report on the manuscript entitled “Research on Liquid Flow Measurement Method Based on Heat Transfer Method” by Qin et al. Manuscript ID: water-2240169
The manuscript is not suitable for publication unless the authors take into consideration the following comments:
1. Page 1 , line 38: The authors wrote “J H Huijsing et al.” It should be written as follows “Huijsing et al.”
1. Page 1, line 41: The authors wrote “He Anding et al.” It should be written as follows “Anding et al.”
2. Page 2, line 44: The authors wrote “Wang Yanjun et al.” It should be written as follows “Wang et al.”
3. Page 2, line 51: The authors wrote “Jiang Zhaoyu et al.” It should be written as follows “Jiang et al.”
4. Page 2, line 59: The authors wrote “Song Chungao et al.” It should be written as follows “Song et al.”
5. Page 2, line 64: The authors wrote “Wang Dongliang et al.” It should be written as follows “Wang et al.”
6. Page 2, line 71: The authors wrote “Klemen Rupnik et al.” It should be written as follows “Rupnik et al.”
7. Page 2, line 89: The authors wrote “Huang Yanlu et al.” It should be written as follows “Huang et al.”
8. Page 2, line 95: The authors wrote “Amina Bekraoui et al.” It should be written as follows “Berkraoui and Hadjadj”
9. Page 2, line 97: The authors wrote “J H Huijsing et al.” It should be written as follows “Huijsing et al.”
2. Page 1, line 38: The authors wrote “Y.M. Yu et al.” It should be written as follows “Yu et al.”
3. Page 1, line 38: The authors wrote “J H Huijsing et al.” It should be written as follows “Huijsing et al.”
10. Page 4, line 159: formula (5):
suddenly appeared. It should be defined in line 157
11. Page 5, line 178: formula (9):
should be T according to formula (3).
12. Page 7, line 238: The authors wrote “
”. It should be written as follows “W/m/
”.
13. Page 8, lines 272-275: The authors should state clearly the type of fluid used in the experiment. Writing “water tank’ in Fig. 5 is not enough.
14. Page 13: Reference (13); line 407: “et al.” should be deleted. The journal named “Oil and Gas Storage and Transportation” has no existence in the international list of the journals shown in the following link: Search - LibGuides at University of Illinois at Urbana-Champaign
15. Page 13: Reference (14); line 409: “et al.” should be deleted. The journal named “Logging Technology” has no existence in the international list of the journals shown in the following link: Search - LibGuides at University of Illinois at Urbana-Champaign
16. Page 13: Reference (15); line 411: “et al.” should be deleted. The journal named “Automation Technology and Application” has no existence in the international list of the journals shown in the following link: Search - LibGuides at University of Illinois at Urbana-Champaign
17. Page 14: Reference (16); line 413: “et al.” should be deleted. The journal named “Petroleum Instruments” has no existence in the international list of the journals shown in the following link: Search - LibGuides at University of Illinois at Urbana-Champaign
18. Page 14: Reference (17); line 415: “et al.” should be deleted. The journal named “Petroleum Pipe and Instrument” has no existence in the international list of the journals shown in the following link: Search - LibGuides at University of Illinois at Urbana-Champaign
1. Page 14: Reference (18); line 417: The names of the authors were written wrongly. The should be written as follows: “Klemen Rupnik, Ivan Bajsić, Jože Kutin” The journal named “Strojniški vestnik-Journal of Mechanical Engineering” has no existence in the international list of the journals shown in the following link: Search - LibGuides at University of Illinois at Urbana-Champaign
2. Page 14: Reference (20); line 421: “et al.” should be deleted. The journal named “Energy Procedia” has no existence in the international list of the journals shown in the following link: Search - LibGuides at University of Illinois at Urbana-Champaign
3. Page 14: Reference (22); line 425: The journal named “Automation and Instrumentation” has no existence in the international list of the journals shown in the following link: Search - LibGuides at University of Illinois at Urbana-Champaign

Author Response
The manuscript is not suitable for publication unless the authors take into consideration the following comments:
1.Page 1 , line 38: The authors wrote “J H Huijsing et al.” It should be written as follows “Huijsing et al.”
The author’s answer: The issues in the text have been corrected.
1.Page 1, line 41: The authors wrote “He Anding et al.” It should be written as follows “Anding et al.”
The author’s answer: The issues in the text have been corrected.
2.Page 2, line 44: The authors wrote “Wang Yanjun et al.” It should be written as follows “Wang et al.”
The author’s answer: The issues in the text have been corrected.
3.Page 2, line 51: The authors wrote “Jiang Zhaoyu et al.” It should be written as follows “Jiang et al.”
The author’s answer: The issues in the text have been corrected.
4.Page 2, line 59: The authors wrote “Song Chungao et al.” It should be written as follows “Song et al.”
The author’s answer: The issues in the text have been corrected.
5.Page 2, line 64: The authors wrote “Wang Dongliang et al.” It should be written as follows “Wang et al.”
The author’s answer: The issues in the text have been corrected.
6.Page 2, line 71: The authors wrote “Klemen Rupnik et al.” It should be written as follows “Rupnik et al.”
The author’s answer: The issues in the text have been corrected.
7.Page 2, line 89: The authors wrote “Huang Yanlu et al.” It should be written as follows “Huang et al.”
The author’s answer: The issues in the text have been corrected.
8.Page 2, line 95: The authors wrote “Amina Bekraoui et al.” It should be written as follows “Berkraoui and Hadjadj”
The author’s answer: The issues in the text have been corrected.
9.Page 2, line 97: The authors wrote “J H Huijsing et al.” It should be written as follows “Huijsing et al.”
The author’s answer: The issues in the text have been corrected.
10.Page 4, line 159: formula (5): suddenly appeared. It should be defined in line 157
The author’s answer: The issues in the text have been corrected.
11.Page 5, line 178: formula (9): should be T according to formula (3).
The author’s answer: The issues in the text have been corrected.
12.Page 7, line 238: The authors wrote “ ”. It should be written as follows “W/m/ ”.
The author’s answer: The issues in the text have been corrected.
13.Page 8, lines 272-275: The authors should state clearly the type of fluid used in the experiment. Writing “water tank’ in Fig. 5 is not enough.
The author’s answer: The main fluid used in this paper is tap water, and a large water tank is used to control the flow. At the same time, the influence of flow pattern on the experimental results is also reduced. Because the volume of the water tank is relatively large, the illustrations in the text do not completely draw the area of the water tank, but just explain the meaning of the water tank.
14.Page 13: Reference (13); line 407: “et al.” should be deleted. The journal named “Oil and Gas Storage and Transportation” has no existence in the international list of the journals shown in the following link: Search - LibGuides at University of Illinois at Urbana-Champaign
The author’s answer: The issues in the text have been corrected.
15.Page 13: Reference (14); line 409: “et al.” should be deleted. The journal named “Logging Technology” has no existence in the international list of the journals shown in the following link: Search - LibGuides at University of Illinois at Urbana-Champaign
The author’s answer: The issues in the text have been corrected.
16.Page 13: Reference (15); line 411: “et al.” should be deleted. The journal named “Automation Technology and Application” has no existence in the international list of the journals shown in the following link: Search - LibGuides at University of Illinois at Urbana-Champaign
The author’s answer: The issues in the text have been corrected.
- Page 14: Reference (16); line 413: “et al.” should be deleted. The journal named “Petroleum Instruments” has no existence in the international list of the journals shown in the following link: Search - LibGuides at University of Illinois at Urbana-Champaign
The author’s answer: The issues in the text have been corrected.
- Page 14: Reference (17); line 415: “et al.” should be deleted. The journal named “Petroleum Pipe and Instrument” has no existence in the international list of the journals shown in the following link: Search - LibGuides at University of Illinois at Urbana-Champaign
The author’s answer: The issues in the text have been corrected.
- Page 14: Reference (18); line 417: The names of the authors were written wrongly. The should be written as follows: “Klemen Rupnik, Ivan Bajsić, Jože Kutin” The journal named “Strojniški vestnik-Journal of Mechanical Engineering” has no existence in the international list of the journals shown in the following link: Search - LibGuides at University of Illinois at Urbana-Champaign
The author’s answer: The issues in the text have been corrected.
2.Page 14: Reference (20); line 421: “et al.” should be deleted. The journal named “Energy Procedia” has no existence in the international list of the journals shown in the following link: Search - LibGuides at University of Illinois at Urbana-Champaign
The author’s answer: The issues in the text have been corrected.
- Page 14: Reference (22); line 425: The journal named “Automation and Instrumentation” has no existence in the international list of the journals shown in the following link: Search - LibGuides at University of Illinois at Urbana-Champaign
The author’s answer: The issues in the text have been corrected.

Round 2
Reviewer 2 Report
The authors did not follow my list of correction as I saw the following :
1) Page 7 line 238: the units of the thermal conductivity were not corrected.
2) The authors did not correct the references in lines 421, 425, 427, 429, and 433 among others mentioned in my Review Report. These journals are not existed in the Western World.
Author Response
Comments and Suggestions for Authors
The authors did not follow my list of correction as I saw the following :
1) Page 7 line 238: the units of the thermal conductivity were not corrected.
The author’s answer: The issues in the text have been corrected.
2) The authors did not correct the references in lines 421, 425, 427, 429, and 433 among others mentioned in my Review Report. These journals are not existed in the Western World.
The author’s answer: The incorrect references in the article have been revised.
